# Modeling Shared Responses in Neuroimaging Studies through MultiView ICA

**Hugo Richard**\*
Inria, Université Paris-Saclay
Saclay, France
`hugo.richard@inria.fr`

**Luigi Gresele**\*
MPI for Intelligent Systems,
MPI for Biological Cybernetics, Tübingen, Germany
`luigi.gresele@tuebingen.mpg.de`

**Aapo Hyvärinen**
Inria, Université Paris-Saclay, Saclay, France
Department of Computer Science HIIT, University of Helsinki, Finland
`aapo.hyvarinen@helsinki.fi`

**Bertrand Thirion**
Inria, Université Paris-Saclay
Saclay, France
`bertrand.thirion@inria.fr`

**Alexandre Gramfort**
Inria, Université Paris-Saclay
Saclay, France
`alexandre.gramfort@inria.fr`

**Pierre Ablin**
Département de Mathématiques et Applications
Ecole Normale Supérieure
Paris, France
`pierre.ablin@ens.fr`

## Abstract

Group studies involving large cohorts of subjects are important to draw general conclusions about brain functional organization. However, the aggregation of data coming from multiple subjects is challenging, since it requires accounting for large variability in anatomy, functional topography and stimulus response across individuals. Data modeling is especially hard for ecologically relevant conditions such as movie watching, where the experimental setup does not imply well-defined cognitive operations. We propose a novel MultiView Independent Component Analysis (ICA) model for group studies, where data from each subject are modeled as a linear combination of shared independent sources plus noise. Contrary to most group-ICA procedures, the likelihood of the model is available in closed form. We develop an alternate quasi-Newton method for maximizing the likelihood, which is robust and converges quickly. We demonstrate the usefulness of our approach first on fMRI data, where our model demonstrates improved sensitivity in identifying common sources among subjects. Moreover, the sources recovered by our model exhibit lower between-session variability than other methods. On magnetoencephalography (MEG) data, our method yields more accurate source localization on phantom data. Applied on 200 subjects from the Cam-CAN dataset it reveals a clear sequence of evoked activity in sensor and source space.

---

# 1 Introduction

The past decade has seen the emergence of two trends in neuroimaging: the collection of massive neuroimaging datasets, containing data from hundreds of participants [66, 69, 64], and the use of naturalistic stimuli to move closer to a real life experience with dynamic and multimodal stimuli [63]. Large scale datasets provide an unprecedented opportunity to assess the generality and validity of neuroscientific findings across subjects, with the potential of offering novel insights on human brain function and useful medical biomarkers. However, when using ecological conditions, such as movie watching or simulated driving, stimulations are difficult to quantify. Consequently the statistical analysis of the data using supervised regression-based approaches is difficult. This has motivated the use of unsupervised learning methods that leverage the availability of data from multiple subjects performing the same experiment; analysis on such large groups boosts statistical power.

Independent component analysis [42] (ICA) is a widely used unsupervised method for neuroimaging studies. It is routinely applied on individual subject electroencephalography (EEG) [47], magnetoencephalography (MEG) [71] or functional MRI (fMRI) [49] data. ICA models a set of signals as the product of a *mixing matrix* and a *source* matrix containing independent components. The identifiability theory of ICA states that having non-Gaussian independent sources is a strong enough condition to recover the model parameters [22]. ICA therefore does not make assumptions about what triggers brain activations in the stimuli, unlike confirmatory approaches like the general linear model [29, 61]. This explains why, in fMRI processing, it is a model of choice when analysing resting state data [5] or when subjects are exposed to natural [48, 4] or complex stimuli such as simulated driving [15]. In M/EEG processing, it is widely used to isolate acquisitions artifacts from neural signal [43], and to identify brain sources of interest [72, 25].

However, unlike with univariate methods, statistical inference about multiple subjects using ICA is not straightforward: so-called group-ICA is the topic of various studies [41]. Several works assume that the subjects share a common mixing matrix, but with different sources [57, 65]. Instead, we focus on a model where the subjects share a common sources matrix, but have different mixing matrices. When the subjects are exposed to the same stimuli, the common source matrix corresponds to the group *shared responses*. Most methods proposed in this framework proceed in two steps [14, 38]. First, the data of individual subjects are aggregated into a single dataset, often resorting to dimension reduction techniques like Principal Component Analysis (PCA). Then, off-the-shelf ICA is applied on the aggregated dataset. This popular method has the advantage of being simple and straightforward to implement since it resorts to customary single-subject ICA method. However, it is not grounded in a principled probabilistic model of the problem, and does not have strong statistical guarantees like asymptotic efficiency.

We propose a novel group ICA method called *MultiView ICA*. It models each subject's dataset as a linear combination of a common sources matrix with additive Gaussian noise. Importantly, we consider that the noise is on the sources and not on the sensors. This greatly simplifies the likelihood of the model which can even be written in closed-form. Despite its simplicity, our model allows for an expressive representation of inter-subject variability through subject-specific functional topographies (mixing matrices) and variability in the individual response (with noise in the source domain). To the best of our knowledge, this is the first time that such a tractable likelihood is proposed for multi-subject ICA. The likelihood formulation shares similarities with the usual ICA likelihood, which allows us to develop a fast and robust alternate quasi-Newton method for its maximization.

**Contribution** In section 2, we introduce the MultiView ICA model, and show that it is identifiable. We then write its likelihood in closed form, and maximize it using an alternate quasi-Newton method. We also provide a sensitivity analysis for MultiView ICA, and show that the choice of the noise parameter in the algorithm has little influence on the output. In section 3, we compare our approach to other group ICA methods. Finally, in section 4, we empirically verify through extensive experiments on fMRI and MEG data that it improves source identification with respect to competing methods, suggesting that the expressiveness and robustness of our model make it a useful tool for multivariate neural signal analysis.

## 2 Multiview ICA for Shared response modelling

**Notation** The absolute value of the determinant of a matrix $W$ is $|W|$. The $\ell_2$ norm of a vector $\mathbf{s}$ is $\|\mathbf{s}\|$. For a scalar valued function $f$ and a vector $\mathbf{s} \in \mathbb{R}^k$, we write $f(\mathbf{s}) = \sum_{j=1}^{k} f(s_j)$ and denote $f'$ the gradient of $f$. *All proofs are deferred to appendix C.*

### 2.1 Model, likelihood and approximation

Given $m$ subjects, we model the data $\mathbf{x}^i \in \mathbb{R}^k$ of subject $i$ as

$$\boxed{\mathbf{x}^i = A^i(\mathbf{s} + \mathbf{n}^i), \;\; i = 1, \ldots, m} \tag{1}$$

where $\mathbf{s} = [s_1, \ldots, s_k]^\top \in \mathbb{R}^k$ are the shared independent sources, $\mathbf{n}^i \in \mathbb{R}^k$ is individual noise, $A^i \in \mathbb{R}^{k \times k}$ are the individual mixing matrices, assumed to be full-rank. We assume that samples are observed i.i.d. For simplicity, we assume that the sources share the same density $d$, so that the independence assumption is $p(\mathbf{s}) = \prod_{j=1}^{k} d(s_j)$. Finally, we assume that the noise is Gaussian decorrelated of variance $\sigma^2$, $\mathbf{n}^i \sim \mathcal{N}(0, \sigma^2 I_k)$, and that the noise is independent across subjects and independent from the sources. The assumption of additive white noise on the sources models individual deviations from the shared sources $\mathbf{s}$. It is equivalent to having noise on the sensors with covariance $\sigma^2 A^i \left(A^i\right)^\top$, i.e. a scaled version of the data covariance without noise.

Since the sources are shared by the subjects, there are many more observed variables than sources in the model: there are $k$ sources, while there are $k \times m$ observations. Therefore, model (1) can be seen as an instance of *undercomplete* ICA. The goal of multiview ICA is to recover the mixing matrices $A^i$ from observations of the $\mathbf{x}^i$. The following proposition extends the standard idenfitiability theory of ICA [22] to multiview ICA, and shows that recovering the sources/mixing matrices is a well-posed problem up to scale and permutation.

**Proposition 1** (Identifiability of MultiView ICA). *Consider* $\mathbf{x}^i, \;\; i = 1 \ldots m$*, generated from* (1)*. Assume that* $\mathbf{x}^i = A'^i(\mathbf{s}' + \mathbf{n}'^i)$ *for some invertible matrices* $A'^i \in \mathbb{R}^{k \times k}$*, independent non-Gaussian sources* $\mathbf{s}' \in \mathbb{R}^k$ *and Gaussian noise* $\mathbf{n}'^i$*. Then, there exists a scale and permutation matrix* $P \in \mathbb{R}^{k \times k}$ *such that for all* $i$*,* $A'^i = A^i P$*.*

We propose a maximum-likelihood approach to estimate the mixing matrices. We denote by $W^i = \left(A^i\right)^{-1}$ the unmixing matrices, and view the likelihood as a function of $W^i$ rather than $A^i$. As shown in Appendix A.1, the negative log-likelihood can be written by integrating over the sources

$$\mathcal{L}(W^1, \ldots, W^m) = -\sum_{i=1}^{m} \log |W^i| - \log \left( \int_{\mathbf{s}} \exp \left( -\frac{1}{2\sigma^2} \sum_{i=1}^{m} \|W^i \mathbf{x}^i - \mathbf{s}\|^2 \right) p(\mathbf{s}) d\mathbf{s} \right), \tag{2}$$

up to additive constants. Since this integral factorizes, i.e. the integrand is a product of functions of $s_j$, we can perform the integration as shown in Appendix A.2. We define a smoothened version of the logarithm of the source density $d$ by convolution with a Gaussian kernel as $f(s) = \log \left( \int \exp(-\frac{m}{2\sigma^2} z^2) d(s-z) dz \right)$ and $\tilde{\mathbf{s}} = \frac{1}{m} \sum_{i=1}^{m} W^i \mathbf{x}^i$ the source estimate. The negative log-likelihood becomes

$$\mathcal{L}(W^1, \ldots, W^m) = -\sum_{i=1}^{m} \log |W^i| + \frac{1}{2\sigma^2} \sum_{i=1}^{m} \|W^i \mathbf{x}^i - \tilde{\mathbf{s}}\|^2 + f(\tilde{\mathbf{s}}). \tag{3}$$

Multiview ICA is then performed by minimizing $\mathcal{L}$, and the estimated shared sources are $\tilde{\mathbf{s}}$. The negative log-likelihood $\mathcal{L}$ is quite simple, and importantly, can be computed easily given the parameters of the model and the data; it does not involve any intractable integral.

For one subject ($m = 1$), $\mathcal{L}(W^1)$ simplifies to the negative log-likelihood of ICA and we recover Infomax [8, 16], where the source log-pdf is replaced with the smoothened $f$.

### 2.2 Alternate quasi-Newton method for MultiView ICA

The parameters of the model are estimated by minimizing $\mathcal{L}$. We propose a combination of quasi-Newton method and alternate minimization for this task. First, $\mathcal{L}$ is non-convex: it is only defined

when the $W^i$ are invertible, which is a non-convex set. Therefore, we only look for local minima as usual in ICA. We propose an alternate minimization scheme, where $\mathcal{L}$ is alternatively diminished with respect to each $W^i$. When all matrices $W^1, \ldots, W^m$ are fixed but one, $W^i$, $\mathcal{L}$ can be rewritten, up to an additive constant

$$\mathcal{L}^i(W^i) = -\log|W^i| + \frac{1-1/m}{2\sigma^2}\|W^i\mathbf{x}^i - \frac{m}{m-1}\tilde{\mathbf{s}}^{-i}\|^2 + f(\frac{1}{m}W^i\mathbf{x}^i + \tilde{\mathbf{s}}^{-i}), \qquad (4)$$

with $\tilde{\mathbf{s}}^{-i} = \frac{1}{m}\sum_{j\neq i} W^j\mathbf{x}^j$. This function has the same structure as the usual maximum-likelihood ICA cost function: it is written $\mathcal{L}^i(W^i) = -\log|W^i| + g(W^i\mathbf{x}^i)$, where $g(\mathbf{y}) = \sum_{j=1}^k f(\frac{y_j}{m} + \tilde{\mathbf{s}}_j^{-i}) + \frac{1-1/m}{2\sigma^2}(y_j - \frac{m}{m-1}\tilde{\mathbf{s}}_j^{-i})^2$. Fast quasi-Newton algorithms [75, 1] have been proposed for minimizing such functions. We employ a similar technique as [75], which we now describe.

Quasi-Newton methods are based on approximations of the Hessian of $\mathcal{L}^i$. The relative gradient (resp. Hessian) [3, 18] of $\mathcal{L}^i$ is defined as the matrix $G^i \in \mathbb{R}^{k\times k}$ (resp. tensor $\mathcal{H}^i \in \mathbb{R}^{k\times k\times k\times k}$) such that as the matrix $E \in \mathbb{R}^{k\times k}$ goes to 0, we have $\mathcal{L}^i((I_k + E)W^i) \simeq \mathcal{L}^i(W^i) + \langle G^i, W^i\rangle + \frac{1}{2}\langle E, \mathcal{H}^i E\rangle$. Standard manipulations yield:

$$G^i = \frac{1}{m}f'(\tilde{\mathbf{s}})(\mathbf{y}^i)^\top + \frac{1-1/m}{\sigma^2}(\mathbf{y}^i - \frac{m}{m-1}\tilde{\mathbf{s}}^{-i})(\mathbf{y}^i)^\top - I_k, \text{ where } \mathbf{y}^i = W^i\mathbf{x}^i \qquad (5)$$

$$\mathcal{H}_{abcd}^i = \delta_{ad}\delta_{bc} + \delta_{ac}\left(\frac{1}{m^2}f''(\tilde{\mathbf{s}}_a) + \frac{1-1/m}{\sigma^2}\right)\mathbf{y}_b^i\mathbf{y}_d^i, \text{ for } a,b,c,d = 1\ldots k \qquad (6)$$

Newton's direction is then $-(\mathcal{H}^i)^{-1}G^i$. However, this Hessian is costly to compute (it has $\simeq k^3$ non-zero coefficients) and invert (it can be seen as a big $k^2 \times k^2$ matrix). Furthermore, to enforce that Newton's direction is a descent direction, the Hessian matrix should be regularized in order to eliminate its negative eigenvalues [53], and $\mathcal{H}^i$ is not guaranteed to be positive definite. These obstacles render the computation of Newton's direction impractical. Luckily, if we assume that the signals in $\mathbf{y}^i$ are independent, severall coefficients cancel, and the Hessian simplifies to the approximation

$$H_{abcd}^i = \delta_{ad}\delta_{bc} + \delta_{ac}\delta_{bd}\Gamma_{ab}^i \text{ with } \Gamma_{ab}^i = \left(\frac{1}{m^2}f''(\tilde{\mathbf{s}}_a) + \frac{1-1/m}{\sigma^2}\right)(\mathbf{y}_b^i)^2. \qquad (7)$$

This approximation is sparse: it only has $k(2k-1)$ non-zero coefficients. In order to better understand the structure of the approximation, we can compute the matrix $(H^iM)$ for $M \in \mathbb{R}^{k\times k}$. We find $(H^iM)_{ab} = \Gamma_{ab}^i M_{ab} + M_{ba}$: $H^iM_{ab}$ only depends on $M_{ab}$ and $M_{ba}$, indicating a simple block diagonal structure of $H^i$. The tensor $H^i$ is therefore easily regularized and inverted: $((H^i)^{-1}M)_{ab} = \frac{\Gamma_{ba}^i M_{ab} - M_{ba}}{\Gamma_{ab}^i \Gamma_{ba}^i - 1}$. Finally, since this approximation is obtained by assuming that the $\mathbf{y}^i$ are independent, the direction $-(H^i)^{-1}G^i$ is close to Newton's direction when the $\mathbf{y}^i$ are close to independence, leading to fast convergence. Algorithm 1 alternates one step of the quasi-Newton method for each subject until convergence. A backtracking line-search is used to ensure that each iteration leads to a decrease of $\mathcal{L}^i$. The algorithm is stopped when maximum norm of the gradients over one pass on each subject is below some tolerance level, indicating that the algorithm is close to a stationary point.

---

**Algorithm 1:** Alternate quasi-Newton method for MultiView ICA

---

**Input:** Dataset $(\mathbf{x}^i)_{i=1}^m$, initial unmixing matrices $W^i$, noise parameter $\sigma$, function $f$, tolerance $\varepsilon$

Set tol$= +\infty$, $\tilde{\mathbf{s}} = \frac{1}{m}\sum_{i=1}^k W^i\mathbf{x}^i$

**while** *tol*$> \varepsilon$ **do**

    tol $= 0$

    **for** $i = 1\ldots m$ **do**

        Compute $\mathbf{y}^i = W^i\mathbf{x}^i$, $\tilde{\mathbf{s}}^{-i} = \tilde{\mathbf{s}} - \frac{1}{m}\mathbf{y}^i$, gradient $G^i$ (eq. (5)) and Hessian $H^i$ (eq. (7))

        Compute the search direction $D = -(H^i)^{-1}G^i$

        Find a step size $\rho$ such that $\mathcal{L}^i((I_k + \rho D)W^i) < \mathcal{L}^i(W^i)$ with line search

        Update $\tilde{\mathbf{s}} = \tilde{\mathbf{s}} + \frac{\rho}{m}DW^i\mathbf{x}^i$, $W^i = (I_k + \rho D)W^i$, tol$= \max($tol$, \|G^i\|)$

    **end**

**end**

**return** *Estimated unmixing matrices $W^i$, estimated shared sources $\tilde{\mathbf{s}}$*

---

### 2.3 Robustness to model misspecification

Algorithm 1 has two hyperparameters: $\sigma$ and the function $f$. The latter is usual for an ICA algorithm, but the former is not. We study the impact of these parameters on the separation capacity of the algorithm, when these parameters do not correspond to those of the generative model (1).

**Proposition 2.** *We consider the cost function $\mathcal{L}$ in eq. (3) with noise parameters $\sigma$ and function $f$. Assume sub-linear growth on $f'$: $|f'(x)| \leq c|x|^{\alpha} + d$ for some $c, d > 0$ and $0 < \alpha < 1$. Assume that $\mathbf{x}^i$ is generated following model (1), with noise parameter $\sigma'$ and density of the source $d'$ which need not be related to $\sigma$ and $f$. Then, there exists a diagonal matrix $\Lambda$ such that $(\Lambda(A^1)^{-1}, \dots, \Lambda(A^m)^{-1})$ is a stationary point of $\mathcal{L}$, that is $G^1, \dots, G^m = 0$ at this point.*

The sub-linear growth of $f'$ is a customary hypothesis in ICA which implies that $d$ has heavier-tails than a Gaussian, and in appendix C.2 we provide other conditions for the result to hold. In this setting, the shared sources estimated by the algorithm are $\tilde{\mathbf{s}} = \Lambda(\mathbf{s} + \frac{1}{m}\sum_{i=1}^{m} \mathbf{n}^i)$, which is a scaled version of the best estimate of the shared sources under the Gaussian noise hypothesis.

This proposition shows that, up to scale, the true unmixing matrices are a stationary point for Algorithm 1: if the algorithm starts at this point it will not move. The question of stability is also interesting: if the algorithm is initialized *close* to the true unmixing matrices, will it converge to the true unmixing matrix? In the appendix C.3, we provide an analysis similar to [17], and derive sufficient numerical conditions for the unmixing matrices to be local minima of $\mathcal{L}$. We also study the practical impact of changing the hyperparameter $\sigma$ on the accuracy of a machine learning pipeline based on MultiviewICA on real fMRI data in the appendix Sec. E.5. As expected from the theoretical study, the performance of the algorithm is barely affected by $\sigma$.

### 2.4 Dimensionality reduction

So far, we have assumed that the dimensionality of each view (subject) and that of the sources is the same. This reflects the standard practice in ICA of having equal number of observations and sources. In practice, however, we might want to estimate fewer sources than there are observations per view; the original dimensionality of the data might in practice not be computationally tractable. The problem of how to perform subject-wise dimensionality reduction in group studies is an interesting one *per se*, and out of the main scope of this work. For our purposes, it can be considered as a preprocessing step for which well-known various solutions can be applied. We discuss this further in section 3 and in appendix F.

## 3 Related Work

Many methods for data-driven multivariate analysis of neuroimaging group studies have been proposed. We summarize the characteristics of some of the most commonly used ones. A more thorough description of these methods can be found in appendix F. For completeness, we start by describing PCA. For a zero-mean data matrix $X$ of size $p \times n$ with $p \leq n$, we denote $X = UDV^\top$ the singular value decomposition of $X$ where $U \in \mathbb{R}^{p \times p}$, $V \in \mathbb{R}^{n \times p}$ are orthogonal and $D$ the diagonal matrix of singular values ordered in decreasing order. The PCA of $X$ with $k$ components is $Y \in \mathbb{R}^{k \times n}$ containing the first $k$ rows of $DV^\top$, and it does not hold in general that $YY^\top = I_k$: for the rest of the paper, what we call PCA does not include whitening of the signals.

**Group ICA** When datasets are high-dimensional, a three steps procedure is often used: first dimensionality reduction is performed on data of each subject separately; then the reduced data are merged into a common representation; finally, an ICA algorithm is applied for shared source extraction. The merging of the reduced data is often done by PCA [13] or multi set CCA [70]. This is a popular method for fMRI [14] and EEG [27] group studies. These methods directly recover only group level, shared sources; when individual sources are needed, additional steps are required (back-projection [13] or dual-regression [6]). In contrast, MultiView ICA finds individual and shared independent components in a single step. Finally, in contrast to the methods described above, our method maximizes a likelihood, which brings statistical guarantees like consistency or asymptotic efficiency. The SR-ICA approach of [73] performs dimension reduction, merging and independent component estimation. It is therefore similar to our method. However, they propose to modify the FastICA algorithm [40] in a rather heuristic way, without specifying an optimization problem, let alone maximizing a likelihood.

In the experiments on fMRI data in appendix E.4, we obtain better performance with MultiView ICA than the reported performance of SR-ICA.

**Likelihood-based models** One can consider the more general model $\mathbf{x}^i = A^i \mathbf{s}^i + \mathbf{n}^i$, where the noise covariance can be learned from the data [34]. The likelihood for this model involves an intractable high dimensional integral that is cumbersome to evaluate, and is then optimized with the Expectation-Maximization (EM) algorithm, which is known to converge slowly and unreliably [10, 56]. Having the simpler model (1) leads to a closed-form likelihood, that can then be optimized by more efficient means than the EM algorithm. In model (1), the noise can be interpreted as individual variability rather than sensor noise. In appendix I, we generate data following model $\mathbf{x}^i = A^i \mathbf{s}^i + \mathbf{n}^i$ and report the reconstruction error. The difference in performance between algorithms is small.

**Structured mixing matrices** One strength of our model is that we only assume that the mixing matrices are invertible and still enjoy identifiability whereas some other approaches impose additional constraints. For instance tensorial methods [7] assume that the mixing matrices are the same up to diagonal scaling. Other methods impose a common mixing matrix [23, 32, 12, 50]. Like PCA, the Shared Response Model [20] (SRM) assumes orthogonality of the mixing matrices. While the model defines a simple likelihood and provides an efficient way to reduce dimension, the SRM model is not identifiable as shown in appendix D, and the orthogonal constraint may not be plausible.

**Matching sources a posteriori** A different path to multi-subject ICA is to extract independent components with individual ICA in each subject and align them. We propose a simple baseline approach to do so called *PermICA*. Inspired by the heuristic of the hyperalignment method [36] we choose a reference subject and first match the sources of all other subjects to the sources of the reference subject. The process is then repeated multiple times, using the average of previously aligned sources as a reference. Finally, group sources are given by the average of all aligned sources. We use the Hungarian algorithm to align pairs of mixing matrices [67]. Alternative approaches involving clustering have also been developed [28, 11].

**Deep Learning** Deep Learning methods, such as convolutional auto-encoders (CAE), can also be used to find the subject specific unmixing [21]. While these nonlinear extensions of the aforementioned methods are interesting, these models are hard to train and interpret. In the experiments on fMRI data in appendix E.4, we obtain better accuracy with MultiView ICA than that of CAE reported in [21].

**Correlated component analysis** Other methods can be used to recover the shared neural responses such as the correlated component approach of Dmochowski [26]. We benchmark our method against its probabilistic version [44] called BCorrCA in Figure 3. Our method yields much better results.

**Autocorrelation** Another way to perform ICA is to leverage spectral diversity of the sources rather than non-Gaussianity. These methods are popular alternative to non-Gaussian ICA in the single-subject setting [68, 9, 58] and they output significantly different sources than non-Gaussian ICA [25]. Extensions to multiview problems have been proposed [46, 24].

## 4 Experiments

All code for the experiments is written in Python. We use Matplotlib for plotting [37] , scikit-learn for machine-learning pipelines [55], MNE for MEG processing [30], Nilearn for fMRI processing and for its CanICA implementation [2], Brainiak [45] for its SRM implementation. In the following, the noise parameter in MultiviewICA is always fixed to $\sigma = 1$. We use the function $f(\cdot) = \log \cosh(\cdot)$, giving the non-linearity $f'(\cdot) = \tanh(\cdot)$. We use the Infomax cost function [8] with the same non-linearity to perform standard ICA, with the Picard algorithm [1] for fast and robust minimization of the cost function. Picard is applied with the default hyper-parameters. The code for MultiViewICA is available online at `https://github.com/hugorichard/multiviewica`.

We compare the following methods to obtain $k$ components: *GroupPCA* is PCA on spatially concatenated data. It corresponds to a transposed version of [62]. *PermICA* is described in the previous section. *SRM* is the algorithm of [20]. *GroupICA* is ICA applied after GroupPCA. *PCA+GroupICA* corresponds to GroupICA applied on subject data that have been first individually reduced by PCA with $k$ components. These two approaches correspond to transposed versions of [12], and are similar to [27]. *CanICA* corresponds to PCA+GroupICA where the merging is done using multi set CCA rather than PCA. The dimension reduction in MultiView ICA and PermICA is performed with SRM

in fMRI experiments and subject-specific PCA in MEG experiments. Initialization is discussed in appendix B. A summary of our quantitative results on real data is available in appendix J.

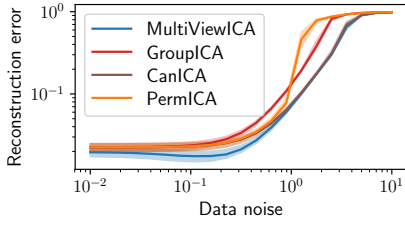

Figure 1: **Synthetic experiment**: reconstruction error of the algorithms on data following model (1).

**Synthetic experiment** We validate our method on synthetic data generated according to the model in equation (1). The sources are generated i.i.d. from a Laplace density $d(x) = \frac{1}{2}\exp(-|x|)$. The mixing matrices $A^1, \cdots, A^m$ are generated with i.i.d. entries following a normal law. Each compared algorithm returns a sequence of estimated unmixing matrices $W^1, \ldots, W^m$. The performance of an algorithm is measured by the reconstruction error between the estimated sources and the true sources. We use $m = 10$ datasets, $k = 15$ sources and $n = 1000$ samples. Each experiment is repeated with 100 random seeds. We vary the noise level in the data generation from $10^{-2}$ to 10.

Multiview ICA has uniformly better performance than the other algorithms, which illustrates the strength of maximum-likelihood based methods. In accordance with results of section 2, it is able to separate the sources even with misspecified noise parameter and source density.

**fMRI data and preprocessing** We evaluate the performance of our approach on four different fMRI datasets. The *sherlock* dataset [19] contains recordings of 16 subjects watching an episode of the BBC TV show "Sherlock" (50 mins). The *forrest* dataset [35] was collected while 19 subjects were listening to an auditory version of the film "Forrest Gump" (110 mins). The *clips* dataset [59] was collected while 12 participants were exposed to short video clips (130 mins). The *raiders* dataset [59] was collected while 11 participants were watching the movie "Raiders of the Lost Ark" (110 mins). The *raiders-full* dataset [59] is an extension of the *raiders* dataset where the first two scenes of the movie are shown twice (130 mins). Like [73], we used full brain data. The rest of the preprocessing is identical to [19]. See E.1 for a detailed description of the datasets and preprocessing steps. Unless stated otherwise we use spatially unsmoothed data, except for the *sherlock* dataset, for which the available data are already preprocessed with a 6 mm spatial smoothing. All datasets are built from successive acquisitions called *runs* that typically last 10 minutes each. We define the chance level as the performance of an algorithm that computes unmixing matrices and projections to lower dimensional space by sampling random numbers from a standard normal distribution.

**Reconstructing the BOLD signal of missing subjects** We want to show that once unmixing matrices have been learned, they can be used to predict evoked responses across subjects, which can be useful to perform transfer learning [74]. We split the data into three groups. First, we randomly choose 80% of all runs from all subjects to form the training set. Then, we randomly choose 80% of subjects and take the remaining 20% runs as testing set. The left-out runs of the remaining 20% subjects form the validation set. The compared algorithms are run on the training set and evaluated using the testing and validation sets. After an algorithm is run on training data, it defines for each subject a *forward operator* that maps individual data to the source space and a *backward operator* that maps the source space to individual data. For instance in ICA the forward operator is the product of the dimensionality reduction projection and unmixing matrix. We estimate the shared responses on the testing set by applying the forward operators on the testing data and averaging. Finally, we reconstruct the individual data from subjects in the validation set by applying the backward operators to the shared responses. We measure the difference between the true signal and the reconstructed one using voxel-wise $R^2$ score. The $R^2$ score between two series $\mathbf{x} \in \mathbb{R}^n$ and $\mathbf{y} \in \mathbb{R}^n$ is defined as $R^2(\mathbf{x},\mathbf{y}) = 1 - \frac{1}{n\,\mathrm{Var}(\mathbf{y})}\sum_{t=1}^{n}(x_t - y_t)^2$, where $\mathrm{Var}(\mathbf{y}) = \frac{1}{n}\sum_{t=1}^{n}(y_t - \frac{1}{n}\sum_{t'=1}^{n}y_{t'})^2$ is the empirical variance of $\mathbf{y}$. The $R^2$ score is always smaller than 1, and equals 1 when $\mathbf{x} = \mathbf{y}$. The experiment is repeated 25 times with random splits to obtain error bars.

In this experiment we apply a 6 mm spatial smoothing to all datasets. The $R^2$ score per voxel depends heavily on which voxel is considered. For example voxels in the visual cortex are better reconstructed in the *sherlock* dataset than in the *forrest* dataset (see Figure 4 in appendix E.2). In Figure 2 (top) we plot the mean $R^2$ score inside a region of interest (ROI) in order to leave out regions where there is no useful information. ROIs are chosen based on the performance of GroupICA (more details in appendix E.2). MultiView ICA has similar or better performance than the other methods on all

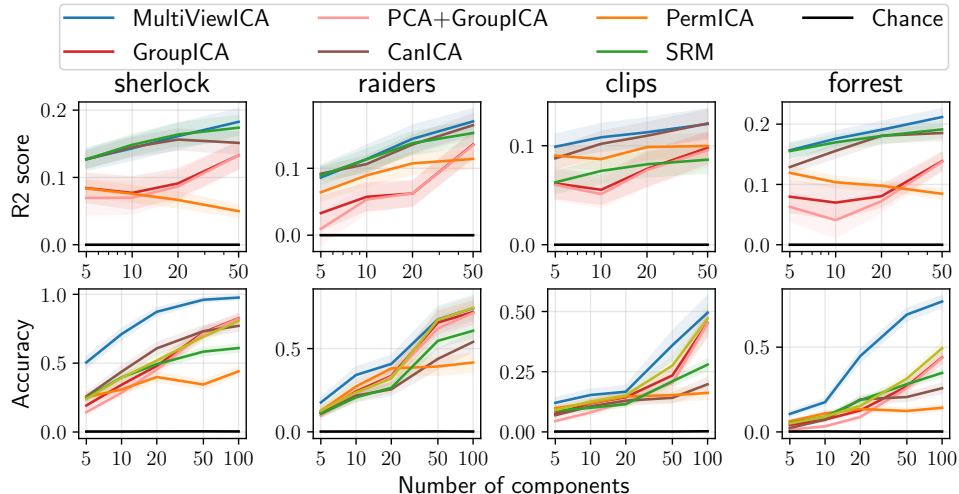

Figure 2: *Top:* **Reconstructing the BOLD signal of missing subjects**. Mean $R^2$ score between reconstructed data and true data (higher is better). *Bottom:* **Between subjects time-segment matching**. Mean classification accuracy. Error bars represent a 95 % confidence interval over cross validation splits.

datasets. This demonstrates its ability to capture inter-subject variability, making it a candidate of choice to handle missing data or perform transfer learning.

**Between subjects time-segment matching** We reproduce the time-segment matching experiment of [20]. We split the runs into a train and test set. After fitting the model on the training set, we apply the forward operator of each subject on the test set yielding individual sources matrices. We estimate the shared responses by averaging the individual sources of each subjects but one. We select a target time-segment (9 consecutive timeframes) in the shared responses and try to localize the corresponding time segment in the sources of the left-out subject using a maximum-correlation classifier. This is a standard evaluation of SRM-like methods also used in [20], [33], [51] or [73]. The time-segment is said to be correctly classified if the correlation between the sample and target time-segment is higher than with any other time-segment (partially overlapping time windows are excluded). We use 5-Fold cross-validation across runs: the training set contains 80% of the runs and the test set 20%, and repeat the experiment using all possible choices for left-out subjects. The mean accuracy is reported in Figure 2 (bottom). MultiView ICA yields a consistent and substantial improvement in accuracy compared to other methods on the four datasets. We see a marked improvement on the datasets sherlock and forrest. A possible explanation lies in the preprocessing pipeline. Sherlock data undergo a 6 mm spatial smoothing and Forrest data are acquired at a higher resolution (7T vs 3T for other data). This affects the signal to noise ratio. In appendix E.5, we compute the accuracy of MultiviewICA on the sherlock dataset with 10 components when the noise parameter varies. MultiviewICA performs consistently well for a wide range of noise parameter values, and only breaks at very high values. It supports the theoretical claim of Prop 2 that the noise parameter is of little importance.

In appendix E.3, we present a variation of this experiment. We measure the ability of each algorithm to extract meaningful shared sources that correlate more when they correspond to the same stimulus than when they correspond to distinct stimuli and show the improved performance of MultiView ICA. In appendix H, we plot the average forward operator across subjects of MultiView ICA and GroupICA with 5 components on the forrest, sherlock, raiders and clips datasets.

**Phantom MEG data** We demonstrate the usefulness of our approach on MEG data. The first experiment uses data collected with a realistic head phantom, which is a plastic device mimicking real electrical brain sources. Eight current dipoles positioned at different locations can be switched on or off. We view each dipole as a subject and therefore have $m = 8$. We only consider the 102 magnetometers. An epoch corresponds to 3 s of MEG signals where a dipole is switched on for 0.4 s with an oscillation at 20 Hz and a peak-to-peak amplitude of 200 nAm. This yields a matrix of size $p \times n$ where $p = 102$ is the number of sensors, and $n$ is the number of time samples. We have access to 100 epochs per dipole. For each dipole, we chose $N_e = 2, \ldots, 16$ epochs at random among our set

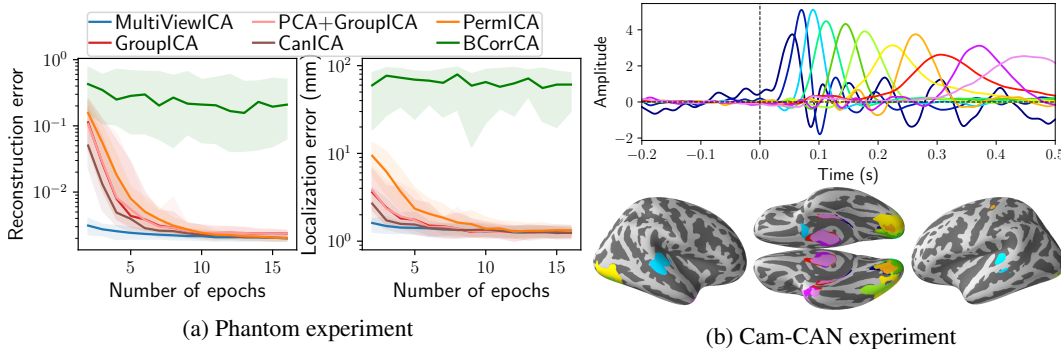

| (a) Phantom experiment | (b) Cam-CAN experiment |

Figure 3: *Left:* **Experiment on MEG Phantom data**. Reconstruction error is the norm of the difference between the estimated and true source. Localization error is the distance between the estimated and true dipole. *Right:* **Experiment on 200 subjects from the CAM-can dataset** *Top:* Time course of 11 shared sources (one color per source). We recover clean evoked potentials. *Bottom:* Associated brain maps, obtained by averaging source estimates registered to a common reference.

of 100 epochs and concatenate them in the temporal dimension. We then apply algorithms on these data to extract $k = 20$ shared sources. As we know the true source (the timecourse of the dipole), we can compute the reconstruction error of each source as the squared norm of the difference between the estimated source and the true source, after normalization to unit variance and fixing the sign. We only retain the source of minimal error. We also estimate for each forward operator the localization of the source by performing dipole fitting using its column corresponding to the source of minimal error. We then compute the distance of the estimated dipole to the true dipole. These metrics are reported in figure 3 when the number of epochs considered $N_e$ varies. We also compare our method to the Bayesian Canonical Correlation Analysis (BCorrCA) of [44]. On this task, BCorrCA is outperformed by ICA methods. MultiView ICA requires fewer epochs to correctly reconstruct and localize the true source.

**Experiment on Cam-CAN dataset** Finally, we apply MultiView ICA on the Cam-CAN dataset [66]. We use the magnetometer data from the MEG of 200 subjects. Each subject is repeatedly presented an audio-visual stimulus. The MEG signal corresponding to these trials are then time-averaged to isolate the evoked response, yielding individual data. The MultiView ICA algorithm is then applied to extract 20 shared sources. 9 sources were found to correspond to noise by visual inspection, and the 11 remaining are displayed in figure 3. We observe that MultiView ICA recovers a very clean sequence of evoked potentials with sharp peaks for early components and slower responses for late components. In order to visualize their localization, we perform source localization for each subject by solving the inverse problem using sLORETA [54], providing a source estimate for each source. Then, we register each source estimate to a common reference brain. Finally, the source estimates are averaged, and thresholded maps are displayed in figure 3. Individual maps corresponding to each source are displayed in appendix G. The figure highlights both early auditory and visual cortices, also suggesting a propagation of the activity towards the ventral regions and higher level visual areas.

## 5 Conclusion

We have proposed a novel unsupervised algorithm that reveals latent sources observed through different views. Using an independence assumption, we have demonstrated that the model is identifiable. In contrast to previous approaches, the proposed model leads to a closed-form likelihood, which we then optimize efficiently using a dedicated alternate quasi-Newton approach. Our approach enjoys the statistical guarantees of maximum-likelihood theory, while still being tractable. We demonstrated the usefulness of MultiView ICA for neuroimaging group studies both on fMRI and MEG data, where it outperforms other methods. In the experiments on fMRI data, we used temporal ICA in order to make use of the fact that subjects were exposed to the same stimuli. However, applying MultiViewICA on transposed data would carry out spatial ICA. Therefore MultiViewICA can be readily used to analyse different kind of neuroimaging data such as resting state data. Our method is not specific to neuroimaging data and could be relevant to other observational sciences like genomics or astrophysics where ICA is already widely used.

## Broader Impact

We develop a novel unsupervised learning method for Independent Component Analysis of a group of subjects sharing commmon sources. Our method is not limited to a particular type of data, and could hence be employed in observational sciences where ICA is relevant: neurosciences, genomics, astrophysics, finance or computer vision for instance. ICA is widely used in these fields as a tool among data processing pipelines, and therefore inherits from all the ethical questions of the fields above. In particular, data collection bias will result in biased outputs. Our algorithm is based on individual linear transforms and therefore decisions based on its application are easier to interpret than more complex models such as deep learning methods: in most applications, the set of parameters has a natural interpretation. For instance in EEG, MEG and fMRI processing, the coefficients of the linear operator can be interpreted as topographic brain maps.

## Acknowledgement and funding disclosure

This work was supported in part by the French government under management of Agence Nationale de la Recherche as part of the "Investissements d'avenir" program, references ANR19-P3IA-0001 (PRAIRIE 3IA Institute) and ANR17-CONV-0003 (DataIA Institute). It has also received funding from the European Union's Horizon 2020 Framework Programme for Research and Innovation under the Specific Grant Agreement No. 945539 (Human Brain Project SGA3), the KARAIB AI chair (ANR-20-CHIA-0025-01) and the European Research Council grant ERC-SLAB-StG-676943. L.G. was hosted for part of this project by the Parietal team at Inria, Saclay, while on an ELLIS exchange. A.H. was additionally supported by CIFAR as a Fellow.

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
