[Supplementary Material]

# APPENDIX

## A  Likelihood

### A.1  Initial form of likelihood

To derive the likelihood, we start by conditioning on $\mathbf{s}$. Then, we make a variable transformation from $\mathbf{x}^i$ to $\mathbf{n}^i = W^i \mathbf{x}^i - \mathbf{s}$, as opposed to the transformation to $\mathbf{s}$ as is usual in ICA. Using the probability transformation formula, we obtain

$$p(\mathbf{x}^i | \mathbf{s}) = |W^i| p_n^i (W^i \mathbf{x}^i - \mathbf{s}) \tag{8}$$

where $p_n^i$ is the distribution of $\mathbf{n}^i$. Note that the $\mathbf{x}^i$ are conditionally independent given $\mathbf{s}$, so we have their joint probability as

$$p(\mathbf{x} | \mathbf{s}) = \prod_{i=1}^{m} |W^i| p_n^i (W^i \mathbf{x}^i - \mathbf{s}) \tag{9}$$

and we next get the joint probability as

$$p(\mathbf{x}, \mathbf{s}) = p(\mathbf{s}) \prod_{i=1}^{m} |W^i| p_n^i (W^i \mathbf{x}^i - \mathbf{s}) \tag{10}$$

Integrating out $\mathbf{s}$ gives Eq. (2).

### A.2  Integrating out the sources

The integral in question, after factorization, is given by

$$\int_{\mathbf{s}} \prod_{j=1}^{k} \exp \left( -\frac{1}{2\sigma^2} \sum_{i=1}^{m} ((\mathbf{w}_j^i)^\top \mathbf{x}^i - s_j)^2 \right) d(s_j) d\mathbf{s} \tag{11}$$

which factorizes for each $j$. Denote $y_j^i = (\mathbf{w}_j^i)^\top \mathbf{x}^i$ and $\tilde{s}_j = \frac{1}{m} \sum_{i=1}^{m} y_j^i$. Fix $j$, and drop it to simplify notation. Then we need to solve the integral

$$\int_s \exp \left( -\frac{1}{2\sigma^2} \sum_{i=1}^{m} (y^i - s)^2 \right) d(s) ds$$

$$= \int_s \exp \left( -\frac{1}{2\sigma^2} [m(\tilde{s} - s)^2 + \sum_{i=1}^{m} (y^i - \tilde{s})^2] \right) d(s) ds$$

$$= \exp \left( -\frac{1}{2\sigma^2} \sum_{i=1}^{m} (y^i - \tilde{s})^2 \right) \int_z \exp \left( -\frac{m}{2\sigma^2} z^2 \right) d(\tilde{s} - z) dz$$

where we have made the change of variable $z = \tilde{s} - s$. The remaining integral simply means that $d$ is smoothed by a Gaussian kernel, which can be computed exactly if $d$ is a Gaussian mixture. We therefore define $f(s) = \log \left( \int_z \exp \left( -\frac{m}{2\sigma^2} z^2 \right) d(s - z) dz \right)$.

## B  Initialization of MultiViewICA

Since the cost function $\mathcal{L}$ is non-convex, having a good initialization can make a difference in the final result. We propose a two stage approach. We begin by applying PermICA on the datasets, which gives us a first set of unimixing matrices $W_1^1, \ldots, W_1^m$. Note that we could also use GroupICA for this task. Next, we perform a diagonal scaling of the mixing matrices, i.e. we find the diagonal matrices $\Lambda^1, \ldots, \Lambda^m$ such that $\mathcal{L}(\Lambda^1 W_1^1, \ldots, \Lambda^m W_1^m)$ is minimized. To do so, we employ Algorithm 1 but only take into account the diagonal of the descent direction at each step: the update rule becomes

$W^i \leftarrow (I_k + \rho \mathrm{Diag}(D))W^i$. The initial unmixing matrices for Algorithm 1 are then taken as $\Lambda^1 W_1^1, \ldots, \Lambda^m W_1^m$.

Empirically, we find that this two stage procedure allows for the algorithm to start close from a satisfactory solution.

## C  Proofs of Section 2

### C.1  Proof of Prop. 1

We fix a subject $i$. Since $\mathbf{s}$ has independent components, so does $\mathbf{s} + \mathbf{n}^i$. Following [22], Theorem 11, there exists a scale-permutation matrix $P^i$ such that $A'^i = A^i P^i$. As a consequence, we have $\mathbf{s} + \mathbf{n}^i = P^i(\mathbf{s}' + \mathbf{n}'^i)$ for all $i$.

Then, we focus on subject 1 and subject $i \neq 1$:

$$\mathbf{s} + \mathbf{n}^1 - (\mathbf{s} + \mathbf{n}^i) = P^1(\mathbf{s}' + \mathbf{n}'^1) - P^i(\mathbf{s}' + \mathbf{n}'^i) \tag{12}$$

$$\mathbf{n}^1 - \mathbf{n}^i = P^1(\mathbf{s}' + \mathbf{n}'^1) - P^i(\mathbf{s}' + \mathbf{n}'^i) \tag{13}$$

$$\iff P^1\mathbf{s}' - P^i\mathbf{s}' = P^i\mathbf{n}'^i - \mathbf{n}^i + \mathbf{n}^1 - P^1\mathbf{n}'^1 \tag{14}$$

Since the right hand side of equation (14) is a linear combination of Gaussian random variables, this would imply that $P^1\mathbf{s}' - P^i\mathbf{s}'$ is also Gaussian. However, given that $\mathbf{s}'$ is assumed to be non-Gaussian, the equality can only hold if $P^1 = P^i$ and both the right and the left hand side vanish. Therefore, the matrices $P^i$ are all equal, and there exists a scale and permutation matrix $P$ such that $A'^i = A^i P$.

### C.2  Proof of Prop. 2

We consider $W^i = \Lambda(A^i)^{-1}$, where $\Lambda$ is a diagonal matrix. We recall $\mathbf{x}^i = A^i(\mathbf{s} + \mathbf{n}^i)$, so that $\mathbf{y}^i = W^i\mathbf{x}^i = \Lambda(\mathbf{s} + \mathbf{n}^i)$. The gradient of $\mathcal{L}$ is given by eq. (5):

$$G^i = \frac{1}{m}f'(\tilde{\mathbf{s}})(\mathbf{s} + \mathbf{n}^i)^\top \Lambda + \frac{1 - 1/m}{\sigma^2}\Lambda\left(\mathbf{n}_i - \frac{1}{m-1}\sum_{j \neq i}\mathbf{n}^j\right)(\mathbf{s} + \mathbf{n}^i)^\top\Lambda - I_k \tag{15}$$

$$= \frac{1}{m}f'(\Lambda(\mathbf{s} + \frac{1}{m}\sum_j \mathbf{n}^j))(\mathbf{s} + \mathbf{n}^i)^\top\Lambda + \frac{\sigma'^2(1 - 1/m)}{\sigma^2}\Lambda^2 - I_k \tag{16}$$

where we write $f'(\mathbf{s}) = \begin{bmatrix} f'(s_1) \\ \vdots \\ f'(s_k) \end{bmatrix}$. Therefore, $G^i$ is diagonal and constant across subjects (because $f'(\Lambda(\mathbf{s} + \frac{1}{m}\sum_j \mathbf{n}^j))(\mathbf{n}^i)^\top = f'(\Lambda(\mathbf{s} + \frac{1}{m}\sum_j \mathbf{n}^j))(\mathbf{n}^{i'})^\top)$. Let us therefore consider only its coefficient $(a, a)$, and let $\lambda = \Lambda_{aa}$:

$$G_{aa}^i = G(\lambda) = \phi(\lambda)\lambda + \frac{\sigma'^2(1 - 1/m)}{\sigma^2}\lambda^2 - 1,$$

where $\phi(\lambda) = \frac{1}{m}f'(\lambda(s_a + \frac{1}{m}\sum_j n_a^j))(s_a + n_a^i)$. One the one hand, we have $G(0) = -1$. On the other hand, if we assume for instance that $f'$ has sub linear growth (i.e. $|f'(x)| \leq c|x|^\alpha + d$ for some $\alpha < 1$) or that $\phi$ is positive, we find that $G(+\infty) = +\infty$. Therefore, $G$ cancels, which concludes the proof.

### C.3  Stability conditions

We consider $W^i = \Lambda(A^i)^{-1}$ where $\Lambda$ is such that the gradients $G^i$ all cancel. We consider a small relative perturbation of $W^i$ of the form $W^i \leftarrow (I_k + E^i)W^i$, and consider the effect on the gradient. We define $\Delta^i = G^i\left((I_k + E^1)W^1, \ldots, (I_k + E^m)W^m\right)$. Denoting $C = \frac{1-1/m}{\sigma^2}$ and

$\tilde{\mathbf{n}} = \frac{1}{m}\sum_{i=1}^{m}\mathbf{n}^i$, we find:

$$\Delta^i = \underbrace{\frac{1}{m}f'\left(\Lambda(\mathbf{s}+\tilde{\mathbf{n}}) + \frac{1}{m}\sum_{j=1}^{m}E^j\Lambda(\mathbf{s}+\mathbf{n}^j)\right)(\mathbf{s}+\mathbf{n}^i)^\top\Lambda(I_k + E^i)^\top + }_{\Delta_1^i} \tag{17}$$

$$\underbrace{C\left(\Lambda\mathbf{n}^i - \frac{1}{m-1}\sum_{j\neq i}\Lambda\mathbf{n}^j + E^i\Lambda(\mathbf{s}+\mathbf{n}^i) - \frac{1}{m-1}\sum_{j\neq i}E^j\Lambda(\mathbf{s}+\mathbf{n}^j)\right)(\mathbf{s}+\mathbf{n}^i)^\top\Lambda(I_k + E^i)^\top}_{\Delta_2^i} \tag{18}$$

$$- I_k \tag{19}$$

$$\tag{20}$$

The first term is expanded at the first order, denoting $S = \sum_{j=1}^{m}E^j$:

$$\Delta_1^i = \frac{1}{m}\left(f'(\Lambda(\mathbf{s}+\tilde{\mathbf{n}})) + f''(\Lambda(\mathbf{s}+\tilde{\mathbf{n}})) \odot \left(\frac{1}{m}\sum_{j=1}^{m}E^j\Lambda(\mathbf{s}+\mathbf{n}^j)\right)\right)(\mathbf{s}+\mathbf{n}^i)^\top\Lambda(I_k+E^i)^\top \tag{21}$$

$$= \frac{1}{m}f'(\Lambda(\mathbf{s}+\tilde{\mathbf{n}}))(\mathbf{s}+\mathbf{n}^i)^\top\Lambda(I_k+E^i)^\top + \frac{1}{m^2}S \odot \left(f''(\Lambda(\mathbf{s}+\tilde{\mathbf{n}}))(\mathbf{s}^2)^\top\Lambda^2\right) \tag{22}$$

$$+ \frac{1}{m^2}E^i \odot \left(f''(\Lambda(\mathbf{s}+\tilde{\mathbf{n}}))((\mathbf{n}^i)^2)^\top\Lambda^2\right) \tag{23}$$

The symbol $\odot$ denotes the element-wise multiplication, $f'(\mathbf{s}) = \begin{bmatrix} f'(s_1) \\ \vdots \\ f'(s_k) \end{bmatrix}$ and $f''(\mathbf{s}) = \begin{bmatrix} f''(s_1) \\ \vdots \\ f''(s_k) \end{bmatrix}$.

Similarly, the second term gives at the first order:

$$\Delta_2^i = \sigma'^2\Lambda^2(I_k+E^i)^\top + (1+\sigma'^2)E^i\Lambda^2 - \frac{1}{m-1}(S-E^i)\Lambda^2 \tag{24}$$

Combining this, we find:

$$\Delta^i = (E^i)^\top + E^i \odot \Gamma^E + S \odot \Gamma^S \tag{25}$$

where

$$\Gamma^E = \left(\frac{1}{m^2}f''(\Lambda(\mathbf{s}+\tilde{\mathbf{n}}))((\mathbf{n}^i)^2)^\top + (1 - \frac{1}{m})\frac{\sigma'^2}{\sigma^2} + \frac{1}{\sigma^2}\right)\Lambda^2$$

$$\Gamma^S = \left(\frac{1}{m^2}f''(\Lambda(\mathbf{s}+\tilde{\mathbf{n}}))(\mathbf{s}^2)^\top - \frac{1}{m\sigma^2}\right)\Lambda^2$$

are $k \times k$ matrices, independent of the subject. This linear operator is the Hessian block corresponding to the $i$-th subject: Denoting $\mathcal{H}$ the Hessian, it is the mapping $\mathcal{H}(E^1, \ldots, E^m) = (\Delta^1, \ldots, \Delta^m)$.

The coefficient $\Delta_{ab}^i$ only depends on $(E_{ab}^i, E_{ba}^i, E_{ab}^1, \ldots, E_{ab}^m)$. Therefore, the Hessian is block diagonal with respect to the blocks of coordinates $(E_{ab}^1, E_{ba}^1, \ldots, E_{ab}^m, E_{ba}^m)$. Denote $\varepsilon = \Gamma_{ab}^E$, $\varepsilon' = \Gamma_{ba}^E$, $\beta = \Gamma_{ab}^S$ and $\beta' = \Gamma_{ba}^S$. The linear operator for the block is:

$$K(\varepsilon, \varepsilon', \beta, \beta') = \left( \begin{array}{cc|cc|c|cc} \varepsilon + \beta & 1 & \beta & 0 & \dots & \beta & 0 \\ 1 & \varepsilon' + \beta' & 0 & \beta' & \dots & 0 & \beta' \\ \hline \beta & 0 & \varepsilon + \beta & 1 & & \beta & 0 \\ 0 & \beta' & 1 & \varepsilon' + \beta' & \ddots & 0 & \beta' \\ \hline \vdots & \vdots & & \ddots & \ddots & \vdots & \vdots \\ \hline \beta & 0 & \beta & 0 & \dots & \varepsilon + \beta & 1 \\ 0 & \beta' & 0 & \beta' & \dots & 1 & \varepsilon' + \beta' \end{array} \right)$$

The positivity of $\mathcal{H}$ is equivalent to the positivity of this operator for all pairs $a, b$. We now assume $\beta\beta' > 0$.

First, we should note that $K(\varepsilon, \varepsilon', \beta, \beta')$ is congruent to $K(\varepsilon\sqrt{\frac{\beta'}{\beta}}, \varepsilon'\sqrt{\frac{\beta}{\beta'}}, \sqrt{\beta\beta'}, \sqrt{\beta\beta'})$ via the basis $\mathrm{diag}((\frac{\beta'}{\beta})^{1/4}, (\frac{\beta}{\beta'})^{1/4}, \cdots, (\frac{\beta'}{\beta})^{1/4}, (\frac{\beta}{\beta'})^{1/4})$. We denote to simplify notation $\alpha = \varepsilon\sqrt{\frac{\beta'}{\beta}}$, $\alpha' = \varepsilon'\sqrt{\frac{\beta}{\beta'}}$ and $\gamma = \sqrt{\beta\beta'}$. We only have to study the positivity of $K(\alpha, \alpha', \gamma, \gamma)$. We have:

$$K(\alpha, \alpha', \gamma, \gamma) = I_m \otimes M_\alpha + \gamma \mathbb{1} \otimes I_2, \quad M_\alpha = \begin{pmatrix} \alpha & 1 \\ 1 & \alpha' \end{pmatrix}$$

Since $I_m \otimes M_\alpha$ and $\gamma\mathbb{1} \otimes I_2$ commute, the minimum value of $\mathrm{Sp}(K)$ is $\min(I_m \otimes M_\alpha) + \min(\gamma\mathrm{Sp}(\mathbb{1})) = \frac{1}{2}(\alpha + \alpha' - \sqrt{(\alpha - \alpha')^2 + 4}) + m\min(0, \gamma)$. Since we assumed $\beta\beta' > 0$ we have $\gamma > 0$. This is similar to the usual ICA case, we find that the condition is $\alpha\alpha' > 1$.

If the following conditions hold for all pair of sources $a, b$, the sources are a local minimum of the cost function:

- $\Gamma_{ab}^S \Gamma_{ba}^S \geq 0$
- $\Gamma_{ab}^E \Gamma_{ba}^E > 1$

# D   Identifiability for Shared Response Model

The shared response model [20] (SRM) models the data $\mathbf{x}^i \in \mathbb{R}^v$ of subject $i$ for $i = 1, \dots, m$ as

$$\mathbf{x}^i = A^i\mathbf{s} + \mathbf{n}^i \text{ with } \mathbf{s} \sim \mathcal{N}(0, \Sigma), \quad \mathbf{n}^i \sim \mathcal{N}(0, \rho_i^2 I_v), \quad {A^i}^\top A^i = I_k$$

where $A^i \in \mathbb{R}^{v,k}$, $\mathbf{s} \in \mathbb{R}^k$ and $\Sigma \in \mathbb{R}^{k,k}$ is a symmetric positive definite matrix.

**Proposition 3.** *SRM is not identifiable*

*Proof.* Let us assume the data $\mathbf{x}^i$ $i = 1, \dots, m$ follow the SRM model with parameters $\Sigma, A^i, \rho_i^2$ $i = 1, \dots, m$.

Let us consider an orthogonal matrix $O \in \mathcal{O}_k$. We call $A'^i = A^i O$ and $\Sigma' = O^\top \Sigma O$. $\Sigma'$ is trivially symmetric positive definite.

Then the data also follows the SRM model with different parameters $\Sigma', A'^i, \rho_i^2$ $i = 1, \dots, m$. □

**Proposition 4.** *We consider the decorrelated SRM model with an additional decorrelation assumption on the shared responses.*

$$\mathbf{x}^i = A^i\mathbf{s} + \mathbf{n}^i \text{ with } \mathbf{s} \sim \mathcal{N}(0, \Sigma), \quad \mathbf{n}^i \sim \mathcal{N}(0, \rho_i^2 I_v), \quad {A^i}^\top A^i = I_k$$

*where $\Sigma$ is a positive* diagonal *matrix. We further assume that the values in $\Sigma$ are all distinct and ranked in ascending order. The decorrelated SRM is identifiable up to sign indeterminacies on the columns of* $\begin{bmatrix} A^1 \\ \vdots \\ A^m \end{bmatrix}$.

*Proof.* The decorrelated SRM model can be written

$$\mathbf{x}^i \sim \mathcal{N}(0, A^i \Sigma A^{i\top} + \rho_i^2 I_v) \ \text{ with } \ A^{i\top} A^i = I_k$$

where $\Sigma$ is a positive diagonal matrix with distincts values ranked in ascending order.

Let us assume the data $\mathbf{x}^i \ \ i = 1, \ldots, m$ follow the decorrelated SRM model with parameters $\Sigma, A^i, \rho_i^2 \ \ i = 1, \ldots, m$. Let us further assume that the data $\mathbf{x}^i \ \ i = 1, \ldots, m$ follow the decorrelated SRM model with an other set of parameters $\Sigma', A'^i, \rho_i'^2 \ \ i = 1, \ldots, m$.

Since the model is Gaussian, we look at the covariances. We have for $i \neq j$

$$\mathbb{E}[\mathbf{x}^i \left(\mathbf{x}^j\right)^\top] = A^i \Sigma A^{j\top} = A'^i \Sigma' A'^{j\top} \ ,$$

The singular value decomposition is unique up to sign flips and permutation. Since eigenvalues are positive and ranked the only indeterminacies left are on the eigenvectors. For each eigenvalue a sign flip can occur simultaneously on the corresponding left and right eigenvector.

Therefore we have $\Sigma' = \Sigma$, $A^i = A'^i D^{ij}$ and $A^j = A'^j D^{ij}$ where $D^{ij} \in \mathbb{R}^{k,k}$ is a diagonal matrix with values in $\{-1, 1\}$. This analysis holds for every $j \neq i$ and therefore $D^{ij} = D$ is the same for all subjects.

We also have for all $i$

$$\mathbb{E}[\mathbf{x}^i \left(\mathbf{x}^i\right)^\top] = A^i \Sigma A^{i\top} + \rho_i^2 I_v = A'^i \Sigma' A'^{i\top} + \rho_i'^2 I_v$$

We therefore conclude $\rho_i'^2 = \rho_i^2, i = 1 \ldots m$.

Note that if the diagonal subject specific noise covariance $\rho_i^2 I_v$ is replaced by any positive definite matrix, the model still enjoys identifiability. $\qquad\square$

# E  fMRI experiments

## E.1  Dataset description and preprocessing

The full brain mask used to select brain regions is available in the Python package associated with the paper.

**Sherlock**  In *sherlock* dataset, 17 participants are watching "Sherlock" BBC TV show (beginning of episode 1). These data are downloaded from `http://arks.princeton.edu/ark:/88435/dsp01nz8062179`. Data were acquired using a 3T scanner with an isotropic spatial resolution of 3 mm. More information including the preprocessing pipeline is available in [19]. Subject 5 is removed because of missing data leaving us with 16 participants. Although *sherlock* data are downloaded as a temporal concatenation of two runs, we split it manually into 4 runs of 395 timeframes and one run of 396 timeframes so that we can perform 5 fold cross-validation in our experiments.

**FORREST**  In FORREST dataset 20 participants are listening to an audio version of the Forrest Gump movie. FORREST data are downloaded from OpenfMRI [60]. Data were acquired using a 7T scanner with an isotropic spatial resolution of 1 mm (see more details in [35]) and resampled to an isotropic spatial resolution of 3 mm. More information about the forrest project can be found at `http://studyforrest.org`. Subject 10 is discarded because not all runs available for other subjects were available for subject 10 at the time of writing. Run 8 is discarded because it is not present in most subjects.

**RAIDERS**  In RAIDERS dataset, 11 participants are watching the movie "Raiders of the lost ark". The RAIDERS dataset belongs to the Individual Brain Charting dataset [59]. Data were acquired using a 3T scanner and resampled to an isotropic spatial resolution of 3 mm. The RAIDERS dataset reproduces the protocol described in [36]. Preprocessing details are described in [59].

Figure 4: **Reconstructing the BOLD signal of missing subjects: Reconstruction R2 score per voxel** We plot for GroupICA, SRM and MultiViewICA, the R2 score per voxel using 50 components for datasets *sherlock*, *forrest*, *raiders* and *clips*. We visually see that data reconstructed by MultiViewICA are more faithful reproduction of the original data than other methods.

**CLIPS** In CLIPS dataset, 12 participants are exposed to short video clips. The CLIPS dataset also belongs to the Individual Brain Charting dataset ([59]). Data were acquired using a 3T scanner and resampled to an isotropic spatial resolution of 3 mm. It reproduces the protocol of original studies described in [52] and [39]. Preprocessing details are described in [59].

At the time of writing, the CLIPS and RAIDERS dataset from the individual brain charting dataset `https://project.inria.fr/IBC/` are available at `https://openneuro.org/datasets/ds002685`. Protocols on the visual stimuli presented are available in a dedicated repository on Github: `https://github.com/hbp-brain-charting/public_protocols`.

### E.2 Reconstructing the BOLD signal of missing subjects: Discussion on ROIs choice

The quality of the reconstructed BOLD signal varies depending on the choice of the region of interest. In Figure 4, we plot for GroupICA, SRM and MultiViewICA, the R2 score per voxel using 50 components for datasets *sherlock*, *forrest*, *raiders* and *clips*. As could be anticipated from the task definition, *forrest* obtains high reconstruction accuracy in the auditory cortices, while *clips* shows good reconstruction in the visual cortex (occipital lobe mostly); the richer *sherlock* and *raiders* datasets yield good reconstructions in both domains, but also in other systems (language, motor). We also see visually see that data reconstructed by MultiViewICA are a better approximation of the original data than other methods. This is particularly obvious for the *clips* datasets where it is clear that voxels in the posterior part of the superior temporal sulcus are better recovered by MultiViewICA than by SRM or GroupICA.

In order to determine the ROIs, we focus on the R2 score per voxel between the BOLD signal reconstructed by GroupICA and the actual bold signal. We run GroupICA with $10, 20$ and $50$ components and select the voxels that obtained a positive R2 score for all sets of components. We discard voxels with an R2 score above 80% as they visually correspond to artefacts and apply a binary opening using a unit cube as the structuring element. The chosen regions are plotted in figure 5.

### E.3 Between-runs time-segment matching

We measure the ability of each algorithm to extract meaningful shared sources that correlate more when they correspond to the same stimulus than when they correspond to distinct stimuli. We use the *raiders-full* dataset, which allows this kind of analysis because subjects watch some selected

Figure 5: **Data-driven choice of ROI** Chosen ROIs for the experiment: Reconstructing the BOLD signal of missing subjects.

Figure 6: **Between runs time-segment matching**. Interesting sources correlates more when they correspond to the same stimulus (same scenes of the movie) than when they correspond to distinct stimuli (different scenes). We extract 20 sources and report the mean accuracy of the 3 best performing sources

scenes from the movie twice, during the first two runs (1 and 2) and the last two (11 and 12). First, the forward operators are learned by fitting each algorithm with 20 components on the data of all 11 subjects using all 12 runs. We then select a subset of 8 subjects and the shared sources are computed by applying the forward operators and averaging. We select a large target time-segment (50 timeframes) taken at random from run 1 and 2, and we try to localize the corresponding sample time-segment from the 10 last runs using a single component of the shared sources. The time-segment is said to be correctly classified if the correlation between the target and corresponding sample time-segment is higher than with any other time-segment (partially overlapping windows are excluded). In contrast to the *between subject time-segment matching* experiment, we obtain one accuracy score per component. We repeat the experiment 10 times with different subsets of subjects randomly chosen and report the mean accuracy of the 3 best performing components in Figure 6. Error bars correspond to a 95 % confidence interval. MultiView ICA achieves the highest accuracy.

We then focus on the 3 best performing components of MultiView ICA. For each component, we plot in Figure 7 (left) the shared sources during two sets of runs where subjects were exposed to the same scenes of the movie. We then study the localisation of these sources. We average the forward operators across subjects and plot the columns corresponding to the components of interest in Figure 7 (right). As each column is seen as a set of weights over all voxels, it represents a spatial map.

The component 1 of the shared responses follows almost the same pattern in the two set of runs corresponding to the same scenes of the movie. The spatial map corresponding to component 1 highlights the language network. In component 2, the temporal patterns during the viewing of

Figure 7: **Between-runs time segment matching: spatial maps and timecourses** *Left:* Time-courses of the 3 shared sources yielding the highest accuracy. The two displayed set of runs correspond to the same scenes in the movie. *Right:* Localisation of the same shared sources in the brain

identical scenes are also very similar. The corresponding spatial map highlights the visual network especially the visual dorsal pathway. In component 3, there exists a similarity however less striking than with the two previous components. The corresponding spatial map highlights a contrast between the spatial attention network and the auditory network.

### E.4 Reproducing time-segment matching experiment

We reproduce the time-segment matching experiments described in [21] and [73] and use two fold classification over runs instead of 5-fold as we have done in the main paper. We used the sherlock data available at `http://arks.princeton.edu/ark:/88435/dsp01nz8062179` and the full brain mask provided in the Python package associated with the paper. We applied high-pass filtering (140 s cutoff) and the time series of each voxel were normalized to zero mean and unit variance.

The results are available in Figure 8.

Figure 8: **Reproducing the time-segment matching experiment of [21, 73]** Mean classification accuracy - error bars represent 95% confidence interval

## E.5 Impact of the hyperparameter $\sigma$

On top of the theoretical guarantees about the robustness of our method to the choice of the $\sigma$ parameter, we investigate its practical impact on the time-matching segment experiment, on the Sherlock dataset with 10 components. We compute the accuracy of the multi-view ICA pipeline with different choice of $\sigma$. This is reported in Fig. 9. The accuracy is constant for a wide range of $\sigma$, only decreasing when $\sigma$ attains very high values.

Figure 9: **Effect of the parameter $\sigma$**: We compute the accuracy of the multiview-ICA pipeline on the time-segment matching experiment for various values of the $\sigma$ hyperparameter over a grid. The accuracy varies only marginally with $\sigma$.

## F Related Work

The following table describes some usual method for extracting shared sources from multiple subjects datasets. The column "Modality/Source" describes the type of data for which each algorithm was *initially* proposed, even though each algorithm could be applied on any type of data. The source type can be either temporal if extracted sources are time courses or spatial if they are spatial patterns.

| Method | Modality/Source | Dimension reduction | Description |
|---|---|---|---|
| SRM [20] | fMRI/Temporal | SRM | The model is $\mathbf{x}^i = A^i\mathbf{s} + \mathbf{n}^i$, with *Gaussian* sources and *orthogonal* mixing matrices $A^i$ |
| GroupPCA [62] | fMRI/Spatial | GroupPCA | A memory efficient implementation of PCA applied on temporally concatenated data. |
| GIFT [13] | fMRI/Spatial | Individual PCA + Group PCA (on component-wise concatenated data) | Single-subject ICA is applied on the aggregated data |
| EEGIFT [27] | EEG/Temporal | Individual PCA + Group PCA (on component-wise concatenated data) | Single-subject ICA is applied on the aggregated data |
| PermICA | Any | Any | Single-subject ICA is applied on each subject's data, and the components are matched using the Hungarian algorithm |
| Clustering approach [28] | fMRI/Spatial | Individual PCA | Single-subject ICA is applied on each subject's data, and the components are matched using a hierarchical clustering algorithm. |
| Measure projection analysis [11] | EEG/Temporal | Individual PCA | Single-subject ICA is applied on each subject's data, and the components are matched using a hierarchical clustering algorithm. |
| TensorICA [7] | fMRI/Spatial | Group PCA (on spatially concatenated data) | TensorICA incorporates ICA assumptions into the PARAFAC model. The mixing matrices $A_1 \cdots A_n$ are such that $A_i = AD_i$ where $A$ is common to all subjects and $D_i$ are subject specific diagonal matrices. |
| Unifying Approach of [34] | fMRI/Spatial | Group PCA (on spatially concatenated data) + GroupPCA (on component-wise concatenated data). | The model is $\mathbf{x}^i = A^i\mathbf{s} + \mathbf{n}^i$ with a Gaussian mixture model on independent sources and a matrix normal prior on the noise. |
| SR-ICA [73] | fMRI/Temporal | SR-ICA | SR-ICA incorporates ICA assumptions into the shared response model. |
| CAE-SRM [21] | fMRI/Temporal | CAE-SRM | A convolutional auto-encoder is used to perform the unmixing. |
| CanICA [70] | fMRI/Spatial | Individual PCA + multi set CCA (on component-wise concatenated data) | CanICA applies single-subject ICA on data reduced with PCA and CCA. |
| Spatial Concat-ICA [65] | fMRI/Spatial | Group PCA (on spatially concatenated data) | ICA is applied on spatially concatenated data. The mixing is constrained to be the same across all subjects. |

| Temporal Concat ICA [23] | EEG/Temporal | Group PCA (on temporally concatenated data) | ICA is applied on temporaly concatenated data. The mixing is constrained to be the same across all subjects. |
| --- | --- | --- | --- |
| coroICA [57] | Any | Any | The model is $\mathbf{x}^i = A\mathbf{s}_i + \mathbf{n}^i$. The mixing is constrained to be the same across all subjects. |

An additional related model is described in [31]. Similarly to our work, the ICA model has noise on the source side. However, the model involves nonlinear mixings, which are computationally unfeasible to optimize via maximum likelihood; a contrastive learning scheme is therefore adopted, and the likelihood is not derived in closed form. No evaluation on neuroimaging datasets is presented.

## G   Detailed Cam-CAN sources

We display each of the 11 shared sources found by Multiview ICA on the Cam-CAN. The time-courses are on the left, the corresponding brain maps are on the right.

## H   Average forward operators on fMRI datasets

We display the average forward operator across subjects on the Raiders, Forrest, Clips and Sherlock datasets obtained with MultiViewICA and GroupICA with 5 components. A 5 mm spatial smoothing was applied on all datasets, and the confound signals corresponding to the 5 components with the highest variance were removed before applying MultiViewICA or GroupICA.

# I    Synthetic benchmark using the model $\mathbf{x}^i = A^i \mathbf{s} + \mathbf{n}^i$

We generate data according to the model $\mathbf{x}^i = A^i \mathbf{s} + \mathbf{n}^i$, where $\mathbf{x}^i \in \mathbb{R}^{50}$, $\mathbf{s} \in \mathbb{R}^{20}$, and $\mathbf{n}^i \sim \mathcal{N}(0, \sigma^2 I_{50})$. After applying individual PCA to obtain signals of dimension 20, we apply the different ICA algorithms and report the reconstruction error in fig. 10.

Figure 10: Synthetic experiment with model $\mathbf{x}^i = A^i \mathbf{s}^i + \mathbf{n}^i$

# J    Summary of our quantitative results

Our quantitative results for the fMRI experiments of time-segment matching and BOLD signal reconstruction and on for the MEG phantom data experiment are summarized, respectively, in Table 2, Table 3 and Table 4. All methods are compared upon extraction of sources with the same dimensionality (20 components).

| Dataset | Method | Accuracy | Confidence interval |
|---------|--------|----------|---------------------|
| clips | Chance | 0.002 | [0.001, 0.003] |
| | CanICA | 0.130 | [0.112, 0.147] |
| | PCA + GroupICA | 0.124 | [0.109, 0.139] |
| | GroupICA | 0.152 | [0.133, 0.171] |
| | PermICA | 0.147 | [0.126, 0.169] |
| | SRM | 0.115 | [0.104, 0.126] |
| | MultiViewICA | **0.167** | [0.142, 0.192] |
| forrest | Chance | 0.002 | [0.001, 0.002] |
| | CanICA | 0.192 | [0.170, 0.214] |
| | PCA + GroupICA | 0.088 | [0.077, 0.098] |
| | GroupICA | 0.154 | [0.137, 0.170] |
| | PermICA | 0.135 | [0.118, 0.152] |
| | SRM | 0.188 | [0.173, 0.203] |
| | MultiViewICA | **0.448** | [0.411, 0.484] |
| raiders | Chance | 0.002 | [0.001, 0.003] |
| | CanICA | 0.256 | [0.220, 0.291] |
| | PCA + GroupICA | 0.331 | [0.289, 0.372] |
| | GroupICA | 0.321 | [0.281, 0.361] |
| | PermICA | 0.381 | [0.341, 0.421] |
| | SRM | 0.265 | [0.240, 0.289] |
| | MultiViewICA | **0.408** | [0.358, 0.458] |
| sherlock | Chance | 0.005 | [0.003, 0.006] |
| | CanICA | 0.607 | [0.567, 0.648] |
| | PCA + GroupICA | 0.454 | [0.416, 0.492] |
| | GroupICA | 0.519 | [0.481, 0.556] |
| | PermICA | 0.399 | [0.365, 0.434] |
| | SRM | 0.493 | [0.465, 0.520] |
| | MultiViewICA | **0.873** | [0.844, 0.903] |

Table 2: Timesegment matching: Summary of our quantitative results. We report the mean accuracy across cross-validation splits.

| Dataset | Method | R2 score | Confidence interval |
|---------|--------|----------|---------------------|
| clips | Chance | 0.000 | [0.000 ,0.000] |
| | CanICA | 0.110 | [ 0.097 , 0.123] |
| | PCA + GroupICA | 0.075 | [ 0.058 , 0.092] |
| | GroupICA | 0.077 | [ 0.059 , 0.094] |
| | PermICA | 0.099 | [ 0.087 , 0.111] |
| | SRM | 0.081 | [ 0.069 , 0.094] |
| | MultiViewICA | **0.114** | [ 0.099 , 0.128] |
| forrest | Chance | 0.000 | [0.000 ,0.000] |
| | CanICA | 0.181 | [ 0.169 , 0.193] |
| | PCA + GroupICA | 0.072 | [ 0.054 , 0.090] |
| | GroupICA | 0.081 | [ 0.062 , 0.099] |
| | PermICA | 0.098 | [ 0.090 , 0.106] |
| | SRM | 0.180 | [ 0.168 , 0.193] |
| | MultiViewICA | **0.191** | [ 0.177 , 0.204] |
| raiders | Chance | 0.000 | [0.000 ,0.000] |
| | CanICA | 0.136 | [ 0.122 , 0.149] |
| | PCA + GroupICA | 0.063 | [ 0.045 , 0.080] |
| | GroupICA | 0.062 | [ 0.043 , 0.081] |
| | PermICA | 0.107 | [ 0.091 , 0.124] |
| | SRM | 0.138 | [ 0.121 , 0.154] |
| | MultiViewICA | **0.144** | [ 0.124 , 0.164] |
| sherlock | Chance | 0.000 | [0.000 ,0.000] |
| | CanICA | 0.156 | [ 0.141 , 0.172] |
| | PCA + GroupICA | 0.087 | [ 0.065 , 0.108] |
| | GroupICA | 0.091 | [ 0.070 , 0.112] |
| | PermICA | 0.067 | [ 0.055 , 0.078] |
| | SRM | **0.164** | [ 0.147 , 0.181] |
| | MultiViewICA | 0.161 | [ 0.142 , 0.180] |

Table 3: Reconstructing the BOLD signal of missing subjects: Summary of our quantitative results. We report the mean R2 score across cross-validation splits.

| Method | Reconstruction error | 1st and 3d quartiles |
|--------|---------------------|----------------------|
| MultiViewICA | **0.0045** | [0.0039, 0.0052] |
| GroupICA | 0.1098 | [0.0549, 0.1734] |
| PCA+GroupICA | 0.1111 | [0.0760, 0.1502] |
| PermICA | 0.0730 | [0.0423, 0.1037] |

Table 4: Phantom MEG data: Summary of our quantitative results with 2 epochs. We report the median reconstruction error across cross-validation splits.