[Reviews · NeurIPS 2020]

Review 1

Summary and Contributions: In this paper the authors propose a novel multi-subject (multi-view) approach for independent components analysis that involves putting a noise term on a set of shared sources, leading to a closed form likelihood term. The authors propose an alternating quasi-Newton method for optimizing their objective function and validate their approach on simulated and real neuroimaging data (MEG and fMRI). Regarding the rebuttal: overall the authors did a reasonable job of addressing the comments, although I do not agree with all the authors claims. Most importantly I think they dismiss the interpretation of the components far too easily

Strengths: Overall this is a reasonably good paper, and my comments are mostly quite minor. The approach is novel to my knowledge and is relatively well described. The evaluation on the real neuroimaging data is also reasonably extensive and well-described. The approach seems to perform well for synthetic data and at least as well as the state of the art for fMRI data, although this is less convincing, e.g. there are no statistical differences between the proposed method and competing methods in terms of R2, although there are in terms of accuracy for time segment matching (although this is not well explained -see below). The algorithm also seems to converge well, which speaks to the benefit of the closed form likelihood.

Weaknesses: I would like to see further exploration of the impact of different parameters (e.g. the noise parameter - currently this is only briefly examined using simulated data). This is important because the authors do not optimise this on the real data experiments. I think the link to existing methods could be better explained (e.g. what is the consequence of putting a noise term on the sources vs. the classical approach). Also, does this lead to differences in the estimated components? this is important since for neuroimaging data analysis (especially fMRI), the components themselves are usually of much greater importance than the reconstruction error (which is usually not even reported in the literature). Also, some of the metrics are not very clear - in particular the time segment matching and Amari distance are not well explained - please define them better here, I don't think it is reasonable to force the reader to dig through existing literature to understand these details.

Correctness: basically correct although I did not go through the appendices in detail.

Clarity: yes, the paper is clearly written subject to the considerations described above.

Relation to Prior Work: Not sufficiently related to existing literature (see above for details).

Reproducibility: Yes

Additional Feedback:


Review 2

Summary and Contributions: This manuscript presents a new ICA approach for group analysis of neuroimaging data. A closed-form solution for optimizing the likelihood is represented and the authors demonstrated the identifiability of their model. The proposed method is benchmarked on several datasets and the results show promising improvement over alternative methods. ***************************** Post Rebuttal Comments ******************************** Thank you for the authors' comprehensive response. However, my recommendations remain the same because: 1) "It supports the theoretical claim of Sec.2.3 that the noise parameter is of little importance": This is more worrying than relaxing. If it has little importance, why do we still keep it in the model? Remember that all contributions of the paper are around changing the model (*) to (1) that is on how we incorporate noise in modeling x. And now we say it has little importance. 2) "noise can be interpreted as individual variability rather than sensor noise": here is exactly where we unrealistically manipulate our interpretation of model components in favor of its scalability and robustness. Despite my serious concerns, I think this is overall a good paper (another wrong but useful model), thus, I would still vote in favor of acceptance.

Strengths: - The paper addresses a relevant and important problem in neuroimaging data analysis. - The technical contributions are robust with enough analytical support. - Several real and synthetic datasets are used in the experiments. - The code is available.

Weaknesses: - All theoretical contributions are based on an unrealistic model in Eq. 1. The proposed model over-siplifies he problem by assuming the noise only on sources and not sensors. - The method section is very dense and difficult to follow. - Some related works (such as DL) are excluded from experimental comparisons. - The alternative methods are selectively used in different experiments. For example, why CanICA is not used in the experiment with synthetic data?

Correctness: All the contributions are based on an unrealistic model assumed in Eq. 1 where the noise is assumed to be only on sources and not sensors. Accordingly, data simulation in experiment on synthetic data follows the same unrealistic assumption (thus no surprise to result in a favorable way). Such a restrictive assumption makes this method just 'another' method with its limitations.

Clarity: The method section is very dense and hard to follow. Otherwise the paper is very well written, the figures are clear and illustrative, and the supplementary material is comprehensive.

Relation to Prior Work: Yes.

Reproducibility: Yes

Additional Feedback: - I suggest adding atleast one paragraph to the text discussing the limitations caused by assuming noise only on sources. - There is little about noise and its parameters in the text while it plays a pivotal role. In all experiments the \sigma is fixed to 1 (this might be the reason we see a jump in Amari ditance when noise goes beyound 1). Is it a reasonable choice? How to select this parameter in data-driven manner? - I would also include the results for CamICA in experiment with experiment data and Phantom MEG.


Review 3

Summary and Contributions: This paper proposes a novel MultiView Independent Component Analysis (ICA) approach to aggregate data from multiple subjects. Unlike other group ICA approaches, the proposed method has a likelihood function solvable in a closed form, and an optimization method for maximizing the likelihood is also introduced.

Strengths: - The paper clearly distinguishes its novelty from previous methods. - The paper proposes an interesting approach to solve a traditional problem. - The paper demonstrates sound experimental results with real neuroimaging data.

Weaknesses: - What is the point of having ``shared'' response modelling? - Considering that the authors demonstrate experiments with real neuroimaging dataset, it would be nice to show what the actual data look like and what the reconstructed signals look like for readers. - It should be better addressed whether the assumption (i.e., having noise on the sources and not sensors) is a more likely scenario. In practice, noise is introduced during the measurement process (i.e., sensors or devices) and not from the nature. - There is no table explaining results in terms of quantitative results. There are various results; however, many of them are turned to supplementary instead of being shown in the main manuscript which makes the paper less convincing. I think the authors should concentrate on explaining some of the outcomes in detail and how their method is advancing scientific discovery. - Also, it also lacks qualitative investigation. Was there any novel finding from neuroscientific perspective?

Correctness: Yes

Clarity: Yes. It is clearly written.

Relation to Prior Work: Yes

Reproducibility: Yes

Additional Feedback: I enjoyed reading this article.


Review 4

Summary and Contributions: I enjoyed reading this contribution. The paper invokes linear ICA-type representations to model shared responses in neuroimaging (fMRI, E/MEG). The main contribution is an analysis and algorithmic advances for a model assuming noisy sources (sources have a common and individual contribution for each subject), while no additive noise in sensor space. The model is evaluated on several real and simulated data sets

Strengths: The main strength is the straightforward generalization of the celebrated Infomax approach to multi-subject data. The specific “noise in source space” assumption enables a closed form likelihood and efficient optimization. The paper is well-written, although much of the content is delegated to appendices. The evaluation involves several simulated and real fMRI and MEG data sets. The algorithm improves in all tests, yet, in most cases modest improvement.

Weaknesses: The fundamental limitation of the work concerns the expressivity of the model. More general factor models and other unsupervised representations would be straightforward to implement in modern probabilistic frameworks, hence, it would be useful to understand the role of several key model assumptions: Linearity, fixed source marginals, and absence of hyperparameter learning. It would be interesting to understand, e.g. how robust the method is to mixed kurtosis source distributions (cf the extended infomax approach often used in EEG processing). The algorithm improves in all tests, yet, mostly modest improvement. Given the title’s focus on shared response, it would have been natural to focus on the robustness of the response rather than on the robustness of the individual mixing matrices (while we note that this is standard in ICA literature). Given the title’s focus on shared neural responses it would have been natural to compare with the other linear decomposition tools developed for exactly this purpose. For example the correlated component approach of Dmochowski et al., 2012 Frontiers in human neuroscience, 6, p.112. Their approach also allows noise in the source space and is robust to individual mixing matrices. The probabilistic version (Kamronn et al., 2015. Neural computation, 27, p.2207) is closely related to so-called Bayesian Group Factor Analysis (Virtanen et al., 2012, AISTATS, p.1269). For additional relevant work see below.

Correctness: The math and algorithm descriptions appear as very careful and code is enclosed.

Clarity: The paper is very well written. Maybe, the conclusions could be improved: With a view to the contribution’s title: Exactly what are the recommendations about the model and the situations in which it should replace conventional approaches.

Relation to Prior Work: While well situated in the literature, comparisons are given with several other approaches. Yet, problematic that correlated components are left out. Also a number of other relevant works could have been noticed (see below comments).

Reproducibility: Yes

Additional Feedback: Line 77: The model assumes full rank (square mixing matrices). This forces some kind of dimension reduction tool (PCA) to be applied to the data prior to ICA. However, it has been shown that such a two-step approach is sup-optimal, relative to algorithms that perform combined dimension reduction and ICA (see: Artoni et al., 2018, NeuroImage 175, p.176). Line 88: The model assumes noise is in source space and independent cross views. Such models have been investigated before in the ICA literature (see e.g.: Lukic et al., 2002, Proceedings International Conference on Image Processing Vol. 2. IEEE). Line 209: What are the implications of fixing the noise variance (\sigma =1)? Line 228: We appreciate that it is easier to test the correctness of the mixing matrices, however, if the focus is on responses, please check if the algorithm leads to better responses. Line 252: Nice evaluation approach! Figure 2: How were hyperparameters optimized for the competing algorithms? Figure 2: “Error bars represent a 95 % confidence interval.” Over which variability/cross-validation?? Figure 2: The accuracy of segment classification is significantly improved for 2 of the 4 data sets – what are the differences between these data sets – while for most other tests the improvements are marginal. Can we understand why data sets “sherlock” and “forest” lead to such marked improvements? **post rebuttal comment Thank you for a clear and comprehensive rebuttal - I have upped my numerical grade.

[Author Response · NeurIPS 2020]

We would like to thank the reviewers for their insightful comments which will help us improve the article. We begin by addressing concerns shared by several reviewers:

*Impact of the noise parameter*: We will add Fig.(a) below in appendix. We compute the accuracy of MultiviewICA on the time-segment matching experiment (Fig.2 in the paper) on the sherlock dataset with 10 components when the noise parameter varies. MultiviewICA performs consistently well for a wide range of noise parameter values, and only breaks at very high values. It supports the theoretical claim of Sec.2.3 that the noise parameter is of little importance.

*Model 1. is unrealistic*: A more natural model is $\mathbf{x}^i = A^i \mathbf{s} + \mathbf{n}^i$ (*) where $\mathbf{n}^i$ is iid. Unfortunately, this model can not be fit in a reasonable amount of time, as reported many times in the literature (see paragraph L.179). For model (1), $\mathbf{x}^i = A^i(\mathbf{s} + \mathbf{n}^i)$, noise can be interpreted as individual variability rather than sensor noise. It offers a way to capture more structured noise, as is often the case in brain signals (cf. ICA based solutions for cleaning EEG or fMRI data). To test its robustness to model mis-specification, we generate data following model (*), and report the reconstruction error in Fig.(b). The difference between algorithms is small. Finally, we argue that whether realistic or not, MultiviewICA is a robust algorithm which in practice improves upon the state-of-the-art on multiple experiments. This discussion will be added in Sec.3.

*Model expressivity, comparison to deep methods* Our method indeed lacks the expressivity of non-linear models. However, linearity is still widespread for brain signal analysis, and non-linear methods are not necessarily better in this setting where the number of samples is limited ([Deep learning for brains?: Different linear and nonlinear scaling in UK Biobank brain images vs. machine-learning datasets, Schulz et al., 2019]). In appendix E.4, we obtain better results than some non-linear deep-learning methods [21]. We will discuss this more thoroughly in Sec.3.

**Rev.1:** *Difference in estimated components* We did not perform qualitative comparison of the estimated components, which is usually difficult due to the intrinsic randomness/non-convexity of ICA. We can only say that the *reconstruction* experiment (Fig.2) and the localization on *MEG phantom data* (Fig.3a) rely only on components, which are hence different. We will add some component maps in the appendix, but leave a more qualitative comparison for future work.

*Unclear experiment/metric*: We will describe the experiments and metric at greater length, and give more intuitions.

**Rev.2:** *Method section is very dense*: We will do our best to unclutter it, and spend more time on important concepts.

*Add CanICA*: We will add CanICA in the synthetic and Phantom experiments (see Figs.(b), (c), (d)).

**Rev.3:** *Shared response?* We will do our best to introduce this key concept more clearly: subjects exposed to the same stimulus should share a common response to it. Many methods have been proposed to reconstruct it (e.g [20]).

*Show data:* We will add in appendix figures showing data, mixing matrices and shared response

*Quantitative and qualitative results:* We will add a table summarizing our quantitative results. The only qualitative result in the paper is the Cam-CAN experiment, which uncovers relevant brain regions and clean evoked potentials. While this is not a discovery, it illustrates the potential of MultiviewICA for unsupervised brain data exploration.

**Rev.4** *Robustness to kurtosis:* The proposed algorithm follows the route of infomax, and therefore only separates super-Gaussian sources. Arguably, most brain sources are super-Gaussian [24]. We plan to develop an extended version which switches between sub-and super-Gaussian density, like extended-Infomax, in a future work.

*Response robustness:* We will add a synthetic experiment where we monitor reconstruction error (Fig.(c)).

*Link with multiview CCA* Thank you for this relevant works which we did not know about. We will mention these methods in Sec.3. We have added Bayesian Multiview CCA (BCorrCA) in the Phantom experiment (Fig.(d) below).

*Dimension reduction*: All algorithms reported here rely on dimension reduction as preprocessing, hence the concerns of Artoni et. al. apply, but we believe that this problem is shared by all methods considered.

*ICA of Lukic et al*: We were not aware of this method. Unlike most ICA methods, it leverages non-stationarity of the sources rather than non-Gaussianity, which might result in very different decompositions [24]. We will mention this work in Sec.3.

*Hyperparameters:* The only hyperparameters of competing algorithms are those of the ICA solver, Picard, that achieves robust convergence by default.

*CV in Fig 2*: The confidence interval is computed over runs and subjects. We will detail this.

*Fig 2 "sherlock" and "forrest" high performance*: Possible explanation: Sherlock data undergo a 6mm spatial smoothing and Forrest data are acquired at a higher resolution (7T vs. 3T for other data). This affects SNR.

(a) Time segment matching experiment with different noise parameter values

(b) Synthetic experiment, model: $x^i = A^i s + n^i$

(c) Synthetic experiment, model: $x^i = A^i(s + n^i)$

(d) Phantom experiment

[Meta-Review · NeurIPS 2020]

The approach is novel and accodring to the reviewers' comments addresses a relevant and important problem in neuroimaging data analysis. Differences to related work are well discussed. Experimental results are sound. The authors have provided a comprehensive response to the reviews. Code for the methods s provided. In sum, this is a solid contribution.